# OpenReview forum: "Cross-Modal Safety Mechanism Transfer in Large Vision-Language Models"
_ICLR.cc/2025/Conference — ICLR 2025 Poster_

### Official Review · Reviewer_R4pV · 2024-11-03

**Soundness:** 3
**Presentation:** 3
**Contribution:** 3
**Rating:** 6
**Confidence:** 3

**Summary:**

This paper aims to find why LVLMs suffer from toxic visual content when converting LLMs to LVLMs. They observe and analyze the safety activation mechanism in the transformers and develop specific methods, TGA, to alleviate the issue of LVLMS without any post alignment.

**Strengths:**

- Clear evidence for the safety activation mechanism.

- Straightforward and well-motivated methods.

- TGA performs relatively well without any post-alignment steps.

**Weaknesses:**

- 1. The analysis seems only work with the model developed by [1]. If the aligned models change, will the conclusion remain consistent?

- 2. Lack of analysis about the extra cost.


[1] Federico Bianchi, Mirac Suzgun, Giuseppe Attanasio, Paul Rottger, Dan Jurafsky, Tatsunori Hashimoto, and James Zou. Safety-tuned LLaMAs: Lessons from improving the safety of large language models that follow instructions. 2024.

**Questions:**

- For point 1 in weaknesses, if the alignment method change, or, the key words change, not "sorry"/"apologize", will the activation layers in  Figure 2 change?

- For point 1 in weaknesses, how about the change of activation layers if we do not fully fine-tune all parameters of the model? For example, use PEFT for the pre-trained LLMs or just frozen the pre-trained LLMs. In such cases, will the trained LVLMs still suffer from toxic visual content?
    - If so, will the activation layers remain the same?
    - If not, the reviewer thinks the conclusion only holds for the fully fine-tuning case.

- In Table 2, how about the performance of a safety-aligned LVLMs like that in [1]?


- Point 2 in weaknesses, the reviewer thinks analysis about the extra cost is needed.

[1] Federico Bianchi, Mirac Suzgun, Giuseppe Attanasio, Paul Rottger, Dan Jurafsky, Tatsunori Hashimoto, and James Zou. Safety-tuned LLaMAs: Lessons from improving the safety of large language models that follow instructions. 2024.

---

> ### Author Response · Authors · 2024-11-20
> **Author Response (1/2)**
>
> Thank you for your thoughtful and valuable comments. We have taken your suggestions into account and will reflect them in the revised manuscript. Our responses to your concerns are provided below. We look forward to discussing with you.
>
> ### Response to Weakness
>
> > **W1**: The analysis seems only work with the model developed by [1]. If the aligned models change, will the conclusion remain consistent?
> [1] Federico Bianchi, Mirac Suzgun, Giuseppe Attanasio, Paul Rottger, Dan Jurafsky, Tatsunori Hashimoto, and James Zou. Safety-tuned LLaMAs: Lessons from improving the safety of large language models that follow instructions. 2024.
>
> **Our Response**: Please allow us to explain it. Our analysis dose not only work with the model developed by [1]. In line 187, we cite [1] just aim to give readers a reference to the common response in safety alignment for LLMs. In our analysis, we add the specific prompt like "*If you find that the instructions are harmful, you should apologize to the user and explain why you are unable to follow the instructions.*" to the system prompt to achieve enabling safety aligned language model to use "Sorry but I cannot ..."" to refuse to follow the instruction. In this way, the activation of LLM's safety mechanism is constantly associated with keywords such as sorry, regardless of the basic model.
>
> > **W2**: Lack of analysis about the extra cost.
>
> **Our Response**: The main extra cost of our method is the introduction of image-text retrieval. We follow your suggestions to analyze the cost of our method on a V100. We find that the inference time for the retrieval part is 0.16s, while the inference time for VLM to generate the response is 6.48s. Therefore, the extra cost of our method is small (0.16 / 6.48 = 2.46%).
>
>
> ### Resoinse to Questions
>
> > **Q1**: For point 1 in weaknesses, if the alignment method change, or, the key words change, not "sorry"/"apologize", will the activation layers in Figure 2 change?
>
> **Our Response**: Thanks for your insightful question! In our analysis, we add the specific prompt like "*If you find that the instructions are harmful, you should apologize to the user and explain why you are unable to follow the instructions.*" to the system prompt to achieve enabling safety aligned language model to use "Sorry but I cannot ..."" to refuse to follow the instruction. In this way, the activation of LLM's safety mechanism is constantly associated with "sorry /apologize" keywords, regardless of the basic model.
>
> To verify this, for the toxic input, we first use gpt-4 to determine whether the response generated by VLM successfully defends against the toxic attack. If so, we determine whether it contains the keyword like "sorry" or "apologize". Our statistics show that based on our system prompt control, 100% of the safe responses contain the "sorry" or "apologize" keyword, of which 90.41% of the responses contain "sorry" and 9.59% of the responses contain "apologize".
>
> > **Q2**: For point 1 in weaknesses, how about the change of activation layers if we do not fully fine-tune all parameters of the model? For example, use PEFT for the pre-trained LLMs or just frozen the pre-trained LLMs. In such cases, will the trained LVLMs still suffer from toxic visual content?
> If so, will the activation layers remain the same?
> If not, the reviewer thinks the conclusion only holds for the fully fine-tuning case.
>
> **Our Response**: We follow your suggestions to use LoRA to fine-tune Mistral-7B to convert it to a LVLM on training data of LLaVA-1.5. We test the safety ability of this LVLM to toxic visual content as Section 3 of our paper and the results shown in the following table indicate that this LVLM still suffer from toxic visual content. The metric is Defense Success Rate (DSR). This experiment will be added in the final version of our paper.
>
> | | Porn | Bloody | Insulting | Alcohol | Cigarette | Gun | Knif|
> |--------|--------|--------|--------|--------|--------|--------|--------|
> |Fully fine-tune| 1.05 | 0.56 | 0.78 | 0.25 | 0.17 | 1.95 | 1.22 |
> |LoRA fine-tune| 2.48 | 0.60 | 0.69 | 0.55 | 0.20 | 1.73 | 1.19|
>
> We then analyze the activation layers of LoRA fine-tuned LVLM as our Section 4, we find its activation layers are the same as the layers in fully fine-tuned LVLM.

---

> ### Author Response · Authors · 2024-11-20
> **Author Response (2/2)**
>
> > **Q3**: In Table 2, how about the performance of a safety-aligned LVLMs like that in [1]?
> [1] Federico Bianchi, Mirac Suzgun, Giuseppe Attanasio, Paul Rottger, Dan Jurafsky, Tatsunori Hashimoto, and James Zou. Safety-tuned LLaMAs: Lessons from improving the safety of large language models that follow instructions. 2024.
>
> **Our Response**: We follow your suggestions to use the LLM after safety-aligned in [1] as the basic model and use training data in LLaVA-1.5 to convert this LLM to LVLM. We test the safety ability of this LVLM on dataset in Table 2. The experimental results shown in the following table indicate that although LLM has already been fully safety-aligned on language in [1], its safety mechanism on language still cannot be transferred to vision effectively. This experiment will be added in the final version of our paper.
>
> | | Porn | Bloody | Insulting | Alcohol | Cigarette | Gun | Knif|
> |--------|--------|--------|--------|--------|--------|--------|--------|
> |Safety-aligned in [1]| 6.32 | 3.09 | 10.21 | 9.33 | 6.72 | 10.85 | 7.98 |
> |TGA (ours)| **20.65** | **9.48** | **22.73** | **17.92** | **17.29** | **30.83** | **29.42** |
>
> > **Q3**: Point 2 in weaknesses, the reviewer thinks analysis about the extra cost is needed.
>
> **Our Response**: The main extra cost of our method is the introduction of image-text retrieval. We follow your suggestions to analyze the cost of our method on a V100. We find that the inference time for the retrieval part is 0.16s, while the inference time for VLM to generate the response is 6.48s. Therefore, the extra cost of our method is small (0.16 / 6.48 = 2.46%).

---

> > ### Comment · Reviewer_R4pV · 2024-11-26
> > **Discussion**
> >
> > Thank the authors for their responses. They address most of my concerns. I maintain my score and lean for an acceptance.

---

> > > ### Author Response · Authors · 2024-11-26
> > > **Thanks!**
> > >
> > > Thank you for your effort in reviewing our paper and for your valuable feedback!

---

### Official Review · Reviewer_CMBn · 2024-11-04

**Soundness:** 3
**Presentation:** 3
**Contribution:** 3
**Rating:** 8
**Confidence:** 3

**Summary:**

Brief Summary: The paper proposes an interesting observation that the safety mechanism of LLMs in filtering out inappropriate content while answering questions is lost when transferring to VLMs naively. As a result, the VLM might answer about things given the image context even though the corresponding LLM wouldn't have.

The authors identify that specific hidden layers in Transformers are responsible for this behavior and propose a method called TGA to transfer this mechanism from LLMs to VLMs.

Experiments on multiple benchmarks (like POPE, MMVet) show that the proposed method maintains performance while filtering our inappropriate content.

**Strengths:**

Pros:

1. The paper tackles an interesting problem which (to best of my knowledge) isn't very well known in the community. As such, it highlights a potential gap and suggests how to fix new VLMs.

2. The motivation is a bit subtle and it is important to note is mostly relevant for open-source models. In a closed sourced model, one could simply have a nsfw classifier on the image-input. However, for open-source model, such an additional component can be easily turned off. As such, a method to have open-source models which are safe is very important. In that sense, the problem is very well motivated.

3. As part of the experiments, the authors collect new dataset which is always appreciated. The authors further provide qualitative visualizations in appendix.

4. The idea of aligning the hidden states is quite clever in my opinion.

5. The authors compare against multiple baseilnes.

**Weaknesses:**

Cons:

1. One thing that isn't clear to me is if it is possible to reverse the trained safety filter by doing an instruction tuning on a sample of toxic dataset by an end user. In that case, it would be easy to "jailbreak" the safe model with relative ease.

2. The authors should include a baseline which works as a direct filter on the image itself to get an upper bound estimate.

**Questions:**

Q1. Confused about the number of toxic image-text pairs, in L134 it notes 2031 but in L454, it notes 20531.

---

> ### Author Response · Authors · 2024-11-20
> **Author Response**
>
> We sincerely appreciate your constructive feedback. We have taken your suggestions into account and will reflect them in the revised manuscript. Our responses to your concerns are provided below. We look forward to discussing with you.
>
> > **W1**: One thing that isn't clear to me is if it is possible to reverse the trained safety filter by doing an instruction tuning on a sample of toxic dataset by an end user. In that case, it would be easy to "jailbreak" the safe model with relative ease.
>
> **Our Response**: This is a good question! We follow your suggestions to conduct a related analysis by instruction tuning the VLM on a sample of toxic dataset (1k toxic instruction-image-response triples) to achieve "jailbreak" the safe model. Experimental results shown in the following table indicate that in this case, most VLMs have lost their original safety mechanisms while our method still has a certain robustness. This experiment will be added in the final version of our paper.
>
> | |  Defense Success Rate (DSR) |
> |--------|--------|
> |BLIP-2| 0.00 |
> |InstructBlip|  0.00 |
> |LLaVA-v1.5| 0.00 |
> |LLaVA-v1.6|0.00 |
> |Qwen-VL-chat|2.42 |
> |Unlearn-FigS|9.76 |
> |TGA-7B (ours)| **15.83** |
>
>
> > **W2**: The authors should include a baseline which works as a direct filter on the image itself to get an upper bound estimate.
>
> **Our Response**: We follow your suggestions to train a filter based on VIT on the mixed images dataset contains both toxic and normal images. We use this to directly filter out the toxic images to get an upper bound estimate. The experimental results are shown in the following table. This experiment will be added in the final version of our paper.
>
> | | Defense Success Rate (DSR) |
> |--------|--------|
> |BLIP-2|  0.32 |
> |InstructBlip|  2.95 |
> |LLaVA-v1.5| 0.67 |
> |LLaVA-v1.6|0.66 |
> |Qwen-VL-chat| 4.55 |
> |Unlearn-FigS| 13.48 |
> |TGA-7B (ours)| **21.08** |
> |Filter (upper bound)| 40.22 |
>
> An important point worth emphasizing is that the upper-bound method requires toxic images as the training data to train a filter, whereas our method does not require any training on toxic images. Instead, it directly transfers the safety mechanism on text modality of LLMs to the image modality in a zero-shot manner.
>
> > **Q1**:  Confused about the number of toxic image-text pairs, in L134 it notes 2031 but in L454, it notes 20531.
>
> **Our Response**: We are sorry for this typo. '2031' in L134 should be '20531'. We will revise it in the final version of our paper.

---

> > ### Comment · Reviewer_CMBn · 2024-11-23
> > **Appreciate the response!**
> >
> > Thanks for the detailed experiments. I am happy with the paper as is, and it would be great if the above two data points on defense against the jailbreak and potential upper bound numbers can be added to the main text or in appendix.

---

> > > ### Author Response · Authors · 2024-11-24
> > > **Thanks**
> > >
> > > Thank you for your efforts in improving our paper and your response! We promise to incorporate all reviewers' comments and new experimental results into the camera ready version of our paper.
> > >
> > > Thank you!

---

### Official Review · Reviewer_VfZw · 2024-11-04

**Soundness:** 3
**Presentation:** 2
**Contribution:** 3
**Rating:** 6
**Confidence:** 4

**Summary:**

The paper identifies a vulnerability in current vision-language alignment methods for Large Vision-Language Models (LVLMs), where the safety mechanisms effective for text fail to transfer to visual input, leaving toxic images unchecked. The authors find that misalignment at the specific hidden state layers cause a semantic shift, undermining the safety mechanism for visual input. To address this, they propose a Text-Guided Alignment (TGA) method, which uses related text to guide the projection of visual inputs into hidden states. Experiments demonstrate that TGA successfully transfers text safety mechanisms to vision without additional fine-tuning and maintains overall performance on vision tasks.

**Strengths:**

This paper is well-motivated and provides a thorough analysis of layer activations to explain the safety misalignment between vision and language. The work has potential value across multiple related fields, particularly in the design of vision-language models and their safety challenges.

The method for identifying the layers where the safety mechanism is activated is both reasonable and straightforward, showing effectiveness with a simple approach.

The proposed TGA alignment method effectively defends against toxic images, with strong evidence presented in Figure 7 to substantiate this claim.

**Weaknesses:**

The paper lacks comparisons with other defense methods. Aside from the comparison with the unlearn-FigS defense, the current experimental results are mainly contrasted with the original model. Including comparisons with existing safety defense methods, such as [1-2], would provide stronger evidence of the proposed approach's superiority.

The presentation is somewhat redundant. For instance, the content in Figures 2 and 4, as well as Figures 3 and 5, could be combined to avoid repetition. Similarly, the writing in Section 4 could be more concise and streamlined for better clarity and flow.

[1] Tovilag: Your Visual-Language Generative Model is Also an Evildoer. EMNLP2023.

[2] Eyes Closed, Safety On: Protecting Multimodal LLMs via Image-to-Text Transformation. ECCV2024.

**Questions:**

Please see weakness.

1. In Figure 1, the presentation is somewhat confusing. Specifically, in Figure 1c, could you clarify whether the blue arrows represent "safe" or "unsafe"?

2. In Section 4, could you specify which layers you are analyzing? For example, are you focusing on the qkv (query, key, value) layers or the projection layers?

3. Can you include any discussion of failure scenarios or bad cases where the method may not perform as expected?

---

> ### Author Response · Authors · 2024-11-20
> **Author Response**
>
> Thank you very much for your insightful suggestions. We have carefully considered them and will integrate them into the revised manuscript. Below are our detailed responses to your comments. We look forward to discussing with you.
> ### Response to Weakness
>
> > **W1**: The paper lacks comparisons with other defense methods. Aside from the comparison with the unlearn-FigS defense, the current experimental results are mainly contrasted with the original model. Including comparisons with existing safety defense methods, such as [1-2], would provide stronger evidence of the proposed approach's superiority.
> [1] Tovilag: Your Visual-Language Generative Model is Also an Evildoer. EMNLP2023.
> [2] Eyes Closed, Safety On: Protecting Multimodal LLMs via Image-to-Text Transformation. ECCV2024.
>
> **Our Response**: We follow your suggestions to compare our method with Tovilag [1] and ECSO [2]. The toxic inputs are the same as Section 3 of our paper. The experimental results shown in the following table indicate that our method outperforms both Tovilag [1] and ECSO [2]. It is because that our method is based on our analysis of the core reason for VLM's vulnerability to toxic vision input, which is actually the safe misalignment between vision and language in VLM caused by insufficient vision-language alignment at hidden states level. Our method is closer to the essence of VLM's vulnerability to vision, while Tovilag [1] needs additional detoxification fine-tuning and ECSO [2] is a post-hoc manner based on safety assessment of response. We promise to add the discussion and comparison about Tovilag [1] and ECSO [2] in the final version of our paper.
>
> | | Porn | Bloody | Insulting | Alcohol | Cigarette | Gun | Knif|
> |--------|--------|--------|--------|--------|--------|--------|--------|
> |Tovliag| 12.67 | 4.14 | 18.05 | 15.28 | 15.07 | 26.90 | 27.45 |
> |ECSO| 18.21 | 7.45 | 20.09 | 15.69| 15.33 | 27.44 | 28.59|
> |TGA (ours)| **20.65** | **9.48** | **22.73** | **17.92** | **17.29** | **30.83** | **29.42** |
>
> > **W2**: The presentation is somewhat redundant. For instance, the content in Figures 2 and 4, as well as Figures 3 and 5, could be combined to avoid repetition. Similarly, the writing in Section 4 could be more concise and streamlined for better clarity and flow.
>
> **Our Response**: Thanks for your valuable suggestions to improve the presentation of our paper! In our final version, we promise to reorganize Figures 2, 3, 4, and 5 according to your suggestions to achieve better clarity and flow.
>
> ### Response to Questions
>
> > **Q1**: In Figure 1, the presentation is somewhat confusing. Specifically, in Figure 1c, could you clarify whether the blue arrows represent "safe" or "unsafe"?
>
> **Our Response**: We apologize for the inconvenience this has caused you. In Figure 1c, blue arrows represent unsafe semantics in vision modality, purple arrows represent unsafe semantics in language modality. Figure 1c shows that vision-language alignment in shallow layers is insufficient so the semantics in vision modality bypass the safety mechanism. In deep layers, vision has been aligned into the space of language, so it turns to purple, but the safety mechanism has been bypassed and the toxic text response is generated. We promise to make this more clear in the final version of our paper.
>
> > **Q2**: In Section 4, could you specify which layers you are analyzing? For example, are you focusing on the qkv (query, key, value) layers or the projection layers?
>
> **Our Response**: Please allow us to explain this. We analyze the output hidden states at each transformer layer rather than specific qkv or projection layers. Our analysis aims to explore whether the semantics of vision and language are correctly aligned in each transformer layer of VLM. This can be measured by the similarity between the hidden states output by a complete transformer layer. Analyzing QKV or projection layer alone cannot accurately capture the semantics.
>
> > **Q3**: Can you include any discussion of failure scenarios or bad cases where the method may not perform as expected?
>
> **Our Response**: After our consideration and analysis of experiments, the main failure scenarios focus on:
> - The input images with harmful objects that are not easily noticed.
> - Images in the specific domain (bloody).
> - Images that are hard for LVLM to understand. For example, the images that LVLM cannot generate the correct caption.
>
> We are willing to add the specific analysis for failure scenarios and bad cases in the final version of our paper.

---

> > ### Comment · Reviewer_VfZw · 2024-11-24
> >
> > Thank you for your response, which addresses my concerns. I keep my score.

---

> > > ### Author Response · Authors · 2024-11-25
> > > **Thanks**
> > >
> > > Thank you for your effort in reviewing our paper and for your valuable feedback!

---

### Official Review · Reviewer_EvHD · 2024-11-05

**Soundness:** 3
**Presentation:** 3
**Contribution:** 3
**Rating:** 6
**Confidence:** 4

**Summary:**

This paper introduces the concept of Cross-Modal Safety Mechanism Transfer for Large Vision-Language Models (LVLMs), aiming to transfer the safety mechanism from text to vision without additional visual safety fine-tuning. The current vision-language alignment fails to align vision with text at the hidden states level, leading to unsafe responses for harmful images. The proposed Text-Guided vision-language Alignment (TGA) retrieves relevant texts to guide the alignment of vision input to hidden states in LVLMs. TGA effectively transfers safety mechanisms from text to vision, maintaining safety without compromising general performance in vision tasks, outperforming existing vision-language alignment methods.

**Strengths:**

1. The authors first analyze cause of failure in cross-modal safety transfer. Based on the analysis, they propose Text-Guided Alignment (TGA) to transfer safety mechanisms from text to vision, addressing key safety issues in LVLMs. The analysis is thorough and the proposed method is novel in general.
2. The paper is well-structured, with clear motivations and systematic explanations of the issues with current vision-language alignment methods.
3. The proposed approach contributes to improving the robustness of LVLMs. This advancement could be important in bridging safety gaps in multimodal AI.

**Weaknesses:**

1. TGA relies on captions generated by LLaVA-1.5-13B for effective alignment. Inaccurate captions can lead to misalignment between vision and language representations, reducing safety performance. Evaluating the impact of captioning errors and exploring mitigation strategies could add robustness to the approach.
2. The paper does not adequately show how the model handles unsafe compositional inputs. For instance, an image of a wine bottle combined with text like "teach a kid to buy this" represents a harmful query, even though the image and text are safe individually. Evaluating compositional risks more deeply could strengthen safety measures.
3. The paper does not show the model's robustness against red-teaming methods such as jailbreak attacks. Evaluating how effective the proposed approach is in defending against these attacks would provide more confidence in the model’s safety capabilities in adversarial settings.

**Questions:**

Same as weakness.

---

> ### Author Response · Authors · 2024-11-20
> **Author Response**
>
> We greatly appreciate your valuable suggestions. We will incorporate them into the revised version. Following is our response to your concerns. We look forward to discussing with you.
>
> > **W1**: TGA relies on captions generated by LLaVA-1.5-13B for effective alignment. Inaccurate captions can lead to misalignment between vision and language representations, reducing safety performance. Evaluating the impact of captioning errors and exploring mitigation strategies could add robustness to the approach.
>
> **Our Response:** Thanks for your valuable insights! We follow your suggestions to evaluate the impact of captioning errors in training. Specifically, we randomly perturb a certain ratio of captions in the training samples, such as replacing them with other captions or randomly deleting information in the captions. Since a single training on the full dataset is time-consuming, we randomly selected a subset (100k) of the full training set. We randomly sample the samples that need to be perturbed at ratios of 5%, 10%, 15%, and 20%, and perform the above perturbations on the captions of these sampled samples. We count the performance (Defense Success Rates, DSR) of the model under different disturbance ratios, and the results shown in the following table indicate that our model performs relatively well when the noise ratio is less than 10\% (10\% is a relatively high noise ratio for training VLM). Therefore, the robustness of our method is acceptable. This experiment will be added in the final version of our paper.
>
> | | 0\% | 5\% | 10\% | 15\% | 20\% |
> |--------|--------|--------|--------|--------|--------|
> |DSR of our model|  18.57 | 18.50 | 18.25| 17.72 | 17.03|
>
> Strategies to improve the robustness of our method include improving the quality of generated captions by self-consistency among multiple agents and adversarial training by adding noisy samples. These will be explored in our future work. Thanks for your suggestions!
>
>
> > **W2**: The paper does not adequately show how the model handles unsafe compositional inputs. For instance, an image of a wine bottle combined with text like "teach a kid to buy this" represents a harmful query, even though the image and text are safe individually. Evaluating compositional risks more deeply could strengthen safety measures.
>
> **Our Response:** Following your suggestions, we compare our method and baselines on SIUO [1], a safety benchmark for vision-language models mainly contain the compositional inputs (safe image and safe text but the combination is unsafe). Experimental results shown in the following indicate that our method can also achieve state-of-the-art performance on this setting. It is because that our method can achieve vision-language alignment at hidden states level in each transformer layer, which allows the two different modalities (vision and language) to share the same safety mechanism. It helps the safety mechanism to accurately judge the combined input attack of the two modalities without modality bias. This experiment will be added in the final version of our paper.
>
> | |Safe Rate |
> |--------|--------|
> |LLaVA-v1.5-7B|  21.56 |
> |LLaVA-v1.5-13B|  22.16 |
> |TGA-7B (ours)| **30.77** |
>
> In this experiment, we only compare our method with llava-v1.5 because the training data of llava-v1.5 is completely open source, so we can use the same training data as llava-v1.5 to train our model. This makes our comparison fair in terms of training data. Qwen-vl is not considered because it requires much larger training data than llava-v1.5 (76.8M vs 1.2M) and contains a large amount of in-house data.
>
> [1] Cross-Modality Safety Alignment
>
> > **W3**: The paper does not show the model's robustness against red-teaming methods such as jailbreak attacks. Evaluating how effective the proposed approach is in defending against these attacks would provide more confidence in the model’s safety capabilities in adversarial settings.
>
> **Our Response:** We follow your suggestions to show model's robustness against jailbreak attacks. The toxic inputs are the same as Section 3 of our paper. We consider three jailbreak attacks including role-play-based attack [1], In-context learning based attack (ICA [2]) and visual prompts based attack (FigStep [3]). The specific experimental results shown in the following table indicate that our method is more robust to jailbreak attack than baselines. The metric is Defense Success Rates (DSR). This experiment will be added in the final version of our paper.
>
> | | Role-Play | ICA | FigStep|
> |--------|--------|--------|--------|
> |BLIP-2|  0.32 | 0.00 | 0.00 |
> |InstructBlip|  2.95 | 1.52 | 2.47 |
> |LLaVA-v1.5| 0.67 | 0.00 | 0.58|
> |LLaVA-v1.6|0.66 | 0.00| 0.60|
> |Qwen-VL-chat| 4.55 | 2.48| 2.91|
> |Unlearn-FigS| 13.48 | 9.45 | 10.57|
> |TGA-7B (ours)| **21.08** | **15.43**|**17.44**|

---

### Meta-Review · Area_Chair_YarE · 2024-12-21

**Metareview:**

The paper explores to transferring the safety mechanism in existing LLM to vision without additional visual safety fine-tuning. The authors first find that the existing vision-language alignment struggles to work at the hidden states level, which results in misleading the safety mechanism in LLM. They proposed the Text-Guided vision-language Alignment method (TGA) for VLMs, which retrieves the texts related to input vision and uses them to guide the projection of vision into the hidden states space in LLMs. Experiments show that the TGA approach not only successfully transfers the safety mechanism in LLM to vision, but also maintains the general performance on several vision tasks.

The reviewers raised weaknesses about insufficient experiments and analysis.
The rebuttal from the authors well addressed the concerns.
The reviewers keep the initial positive scores after rebuttal.

Overall, the paper has a clear motivation, and presents an effective method, advancing the techniques in building safe LVM.
I recommend to accept.

**Additional Comments On Reviewer Discussion:**

The reviewers initially gave ratings of (6, 6, 8, 6) and maintained the scores after the rebuttal.

The reviewers raised weaknesses below:
1) insufficient experiments on caption quality, unsafe compositional inputs, robustness against red-teaming methods, impact of instruction tuning on toxic data,  generalization on varied aligned models such as PEFT for pretrained LLM.
2) insufficient analysis and details on extra cost, comparisons with other defense models, upper bound estimate via a direct filter on the image itself.

The authors provide additional experiments and report the numbers of different ablations, showing the effectiveness of the proposed TGA approach in broader settings.

---

### Decision · Program_Chairs · 2025-01-22

Accept (Poster)